# LLavaCode: Compressed Code Representations for Retrieval-Augmented Code Generation

## Abstract

Retrieval-augmented generation has emerged as one of the most effective approaches for code completion, especially when context from the surrounding repository is important. However, adding this context substantially increases sequence length, which slows inference—an important limitation for interactive settings such as IDEs. In this work, we introduce LlavaCode, a framework that compresses context into compact, semantically rich representations that remain interpretable to code LLMs. This improves generation quality while reducing prompt augmentation to only a few compressed single-token vectors. Our approach requires training only a small projector module and introduces negligible additional latency, yet it significantly improves the prediction quality of code LLMs. Our experiments show that LlavaCode enables a 20–38% reduction in Time-to-First-Token (TTFT) on line-completion tasks compared with uncompressed RAG.

## 1 Introduction

Recently, more and more IDEs started to feature code completion as one of the central tools. Code editors such as Windsurf[1] and Cursor[2] started integrating large language models (LLMs) to provide single- and multiline prediction, which substantially improve developer productivity, but they also impose strict latency requirements: even small delays in time-to-first-token (TTFT) break the interactive coding experience and using this feature becomes frustrating.

Additionally, RAG, a retrieval-augmented generation method (Lewis et al., 2021), has been widely adopted to improve both QA and completion quality, since it allows models to incorporate external context such as documentation, relevant snippets of code or function declarations into the prompt (Figure 1a). However, the additional tokens from retrieval significantly increase prompt processing time and, consequently, TTFT, making vanilla RAG less practical for latency-critical settings like code completion.

A promising solution is context compression via embedding projection. Originally introduced in multimodal models such as Flamingo (Alayrac et al., 2022) and LLaVA (Liu et al., 2023), these methods use a separate visual encoder and a lightweight projection module to map input image embeddings into a small set of tokens for the language model. Subsequent works, such as xRAG (Cheng et al., 2024), extended this idea to textual retrieval, showing that compressed representations can match vanilla RAG performance while reducing inference cost.

Despite this progress, no prior work has applied embedding projection to the code completion task, where the latency–quality trade-off is especially severe. Furthermore, existing training objectives (e.g., cross-entropy) are poorly aligned with developer-relevant code generation quality metrics such as Exact Match (EM) and Edit Similarity (ES), limiting the effectiveness of current approaches. Additionally, we can incorporate other code modalities, such as Abstract Syntax Trees (AST), into the retrieved embeddings to enrich the representations with syntactic information.

In this work, we address both challenges. We introduce LlavaCode—a LLaVA-style projection mechanism that incorporates retrieved context into the model's input while adding only about 10 tokens to the prompt length. The projector is trained using our three-component composite loss:

---

[1]Windsurf homepage
[2]Cursor homepage

cross-entropy, an RL-based term that directly optimizes EM and ES, and a novel cosine-alignment loss that preserves distinctions in the compressed representations.

Our contributions are the following:

- To the best of our knowledge, our approach is the first to apply LLaVA-like embedding projection to code completion tasks *without* embedder or LLM finetuning, resulting in higher quality scores with negligible latency increase compared to base model, while maintaining 20-38% better latency compared to full RAG.

- Prior projection-training methods—whether based solely on cross-entropy or on cross-entropy combined with auxiliary losses—proved insufficient for code completion. To address this, we designed a composite loss that integrates cross-entropy, an RL-inspired component, and a novel cosine-alignment term that preserves distinctions in the compressed representations.

- We've experimented with incorporating additional code modalities such as ASTs to investigate whether alternative representations of code can improve representation quality.

All the code and weights for projector modules will be available under permissive license.

## 2 RELATED WORK

### 2.1 CODING LLMS

StarCoder (Li et al., 2023) introduced a family of code generation models, including larger LLMs optimized for code-centric dialogue and smaller ones tailored for code completion. Trained on the permissively licensed The Stack dataset (Kocetkov et al., 2022), these models achieved strong performance, surpassing most prior approaches on both code completion and instruction-following benchmarks. The Qwen-2.5-Coder series (Hui et al., 2024) represented another significant advancement in code-focused LLMs. Trained on a proprietary mixture of data, the models were released in sizes ranging from 0.5B to 32B parameters and were designed to support text completion, code chat, and fill-in-the-middle tasks.

### 2.2 CONTEXT COMPRESSION METHODS

Despite decoder-only transformer optimizations such as KV-Caching (Pope et al., 2022) and more efficient attention implementations like GQA (Ainslie et al., 2023), time per-token inference latency still scales linearly with context size. Since Retrieval Augmented Generation (Lewis et al., 2021) retrieves information from the knowledge base and puts it into the context of language models, this increases the context size that needs to be processed and subsequently increases end-to-end latency.

In the paper xRAG (Cheng et al., 2024) the authors propose an approach, which is similar to multi-modal language models training: they push the embedding vector of the retrieved text from textual encoder through a lightweight projector layer to align it with the reader model. The resulting architecture is trained in a two-stage manner. In the first stage, both the encoder and LLM are frozen, while the projection layer is trained with cross-entropy loss on paraphrases of the same document. During the second stage, the projector is trained on a mix of tasks such as reading comprehension, open-domain QA and summarization, adding self-distillation from RAG teacher via KL term alongside with usual negative log-likelihood loss. Models trained in such way perform competitively with vanilla RAG systems, while being much more efficient and having lower TTFT due to the reduction in prompt length.

Our approach is conceptually similar to xRAG method. By using a LLaVA-like projection from the encoder to the code completion model, we compress the retrieved context and maintain good generation quality, while lowering the TTFT. However, due to the specificity of our domain, we applied additional techniques to increase code-specific metrics and quality of predictions. Furthermore, we train only the projector with both the encoder and reader LLM frozen in a single stage manner.

## 2.3 Embedding models for code

Code-search embeddings are commonly obtained by converting a decoder-only language model into embedding model by training them to produce last token embeddings for code search via contrastive learning. One such model is Qwen3-Embedding-0.6B (Zhang et al., 2025), which was converted from Qwen3-0.6B (Yang et al., 2025) model. Initialized from a powerful pretrained decoder-only model, Qwen3-Embedding-0.6B shows competitive scores on MTEB (Muennighoff et al., 2023) benchmarks among similarly sized embedding models.

Additionally, some of the encoder models were trained not only on pure text and code data, but also on structured graphs, retrieved from code, such as Data Flow Graphs (DFG) and Abstract Syntax Trees (AST). Examples of such models are GraphCodeBERT (Guo et al., 2021) and UniX-coder (Guo et al., 2022) models, which joined both code, text and graph data to improve representation quality for code-understanding and retrieval tasks.

We have evaluated representative models as encoders in our architecture to investigate how different modalities of code effect the projection quality.

## 2.4 Reinforcement Learning in Language Modeling

Training language models solely for next-token prediction optimizes perplexity but not other objectives such as lack of toxicity, aligning with human preferences, or – specifically for our task – Exact Match (EM) and Edit Similarity (ES) scores.

In the Self-Critical Sequence Training (SCST) paper (Rennie et al., 2017), a variation of REIN-FORCE (Williams, 1992) with a baseline is applied to train an image captioning model. SCST uses the reward of the sequence produced by the current model under the test-time inference algorithm as the baseline, yielding an unbiased, lower-variance REINFORCE estimator.

In our work, we utilize the same REINFORCE-like approach as in SCST, but without baseline term. We directly optimize $ES + EM$ metric, which leads to performance increase.

# 3 Methodology

## 3.1 Model architecture

To decrease the amount of tokens in the context of RAG reader model, we need to somehow compress the retrieved information. In case of LlavaCode, we compress retrieved chunks of code using an off-the-shelf embedding models and then use a small LLaVA-like projector to make it align better with the embeddings of the reader model.

To compress the retrieved context, we use embedding model, which transforms a chunk of code into a single embedding vector. This single vector is being passed through a projection layer, which converts this embedding into a shape that is compatible with LLM embeddings. In our experiments, we take top-10 retrieved chunks per completion and compress them into 10 embeddings, which are concatenated with the LLM embedding of the prompt (Figure 1b). This leads to negligible latency increase (see Section 5 for more latency measurements), since we directly retrieve precomputed projections from the RAG database, without the need to inference the encoder model and the projector module at the time of code completion.

For our experiments, we use Qwen-2.5-Coder family of models as code-completion LLMs and Qwen-3-Embedding-0.6B (Zhang et al., 2025) or UnixCoder (Guo et al., 2022) as encoders. The projector follows the same architecture as projector of LLaVA (Liu et al., 2023): an MLP, with GeLU (Hendrycks & Gimpel, 2023) activation function and a LayerNorm (Ba et al., 2016). We selected two MLP architectures: a 2-layer and a 3-layer MLP. For more details on projector architecture see Table 3 and Appendix C.

## 3.2 Cross-Entropy Issue

In previous works, both in textual and multimodal compression, training was carried out in two stages. On the first stage, the projection layer is pretrained on a simple task to enable better alignment

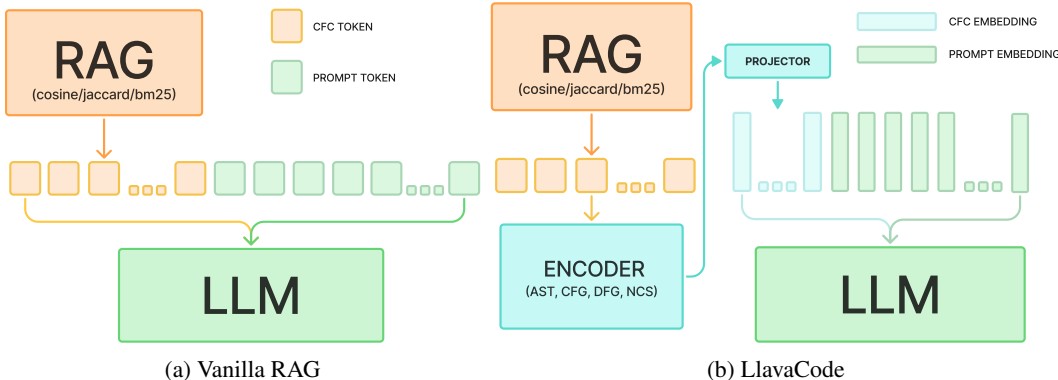

(a) Vanilla RAG                    (b) LlavaCode

Figure 1: Comparison between Vanilla RAG 1a and LlavaCode 1b architectures. Instead of retrieving text passages and putting them into the context of the reader language model, LlavaCode uses a pretrained encoder to compress the text representations and projects them into continuous tokens, thus, reducing the prompt processing time.

of the compressed embeddings with a large language model. On this stage the model is frozen and only projection layer is trained. On the second stage, either the model and the projector or only the projector are trained on downstream tasks. In contrast, our approach uses a single-stage training, omitting the pretraining stage. We've experimented with pretraining the projector, but this did not yield any improvements. More information on pretraining is available in Appendix E.

Most prior work on the related task—training a projection from encoder outputs into the embedding space of an LLM — has relied on instruction-tuned models and QA datasets, and trained primarily with cross-entropy loss (Jaegle et al., 2021; Liu et al., 2023; Zemskova & Yudin, 2025). There are, however, notable exceptions. For example, xRAG incorporated KL divergence loss (Cheng et al., 2024), reporting that it had a greater impact on downstream performance than NLL loss. Another deviation from pure cross-entropy training is Flamingo (Alayrac et al., 2022), which employed the two-term contrastive loss introduced in Radford et al. (2021).

Cross-entropy (negative log-likelihood) is the standard objective for training autoregressive LLMs: it measures how well the model's predicted next-token distribution matches the target distribution. The formula for cross-entropy loss is the following:

$$\mathcal{L}_{CE}(\theta) = -\frac{1}{T}\sum_{t=1}^{T}\log p_\theta(y_t|y_1,\ldots,y_{t-1}).$$

In our experiments, we have found that relying solely on cross-entropy loss was insufficient, since it does not directly correlate with EM and ES metrics. Exact Match measures the percentage of predictions that match the reference output exactly, character for character. It is a strict metric that gives credit only for completely correct generations. Edit Similarity measures the similarity between the prediction and the reference based on the minimum number of edits (insertions, deletions, substitutions) needed to transform one into the other. It provides a softer evaluation by rewarding partial correctness. These are sequence-length metrics, whereas cross-entropy is token-level, maximizing the likelihood of the next token prediction given ground truth. Therefore, it is to be expected that optimizing only for cross-entropy led to suboptimal results on key target metrics — EM and ES — even though it produced lower cross-entropy loss value compared to the baseline model (see ablation study in Table 1). These results motivated us to explore methods for directly optimizing sequence-based metrics, including approaches from reinforcement learning.

### 3.3 REINFORCE

As noted in Rennie et al. (2017), deep generative models for text are typically trained to maximize the likelihood of the next ground-truth word conditioned on the previous ground-truth word via backpropagation. This training paradigm is commonly referred to as Teacher Forcing (Bengio

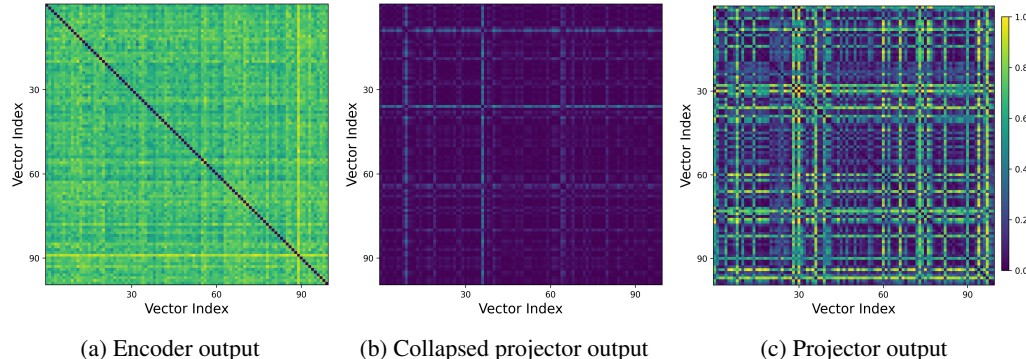

(a) Encoder output        (b) Collapsed projector output        (c) Projector output

Figure 2: Pairwise cosine distances between vector outputs. While the encoder representations remain well-separated (a), the projected vectors may collapse, becoming nearly indistinguishable (b). Introducing the Cosine Alignment Loss 3 helps preserve the distinctions among the projections, preventing excessive overlap.

et al., 2015). However, it introduces a discrepancy between training and inference: at test time, the model generates each word conditioned on its own previous predictions rather than the ground-truth sequence. This exposure bias (Ranzato et al., 2015) can lead to the accumulation of errors during generation, as the model has never been exposed to its own predictions during training.

Our target metrics — Exact Match (EM) and Edit Similarity (ES) — are inherently affected by teacher-forcing bias, as they evaluate predictions at the sequence level. Previous studies have shown that both exposure bias and the non-differentiability of sequence-based evaluation metrics can be mitigated using techniques from Reinforcement Learning (RL) (Sutton & Barto, 1998). In particular, Ranzato et al. (2015) and Rennie et al. (2017) apply the REINFORCE algorithm (Williams, 1992) to directly optimize non-differentiable, sequence-level metrics.

Assume we are training an LLM decoder model with parameters $\theta$. REINFORCE is based on the observation that the expected gradient of a non-differentiable reward function can be computed as follows:

$$\nabla_\theta \mathcal{L}_R(\theta) = -\mathbb{E}_{y \sim p_\theta} \left[ r(y) \, \nabla_\theta \log p_\theta(y) \right], \tag{1}$$

where $y = (y_1, \ldots, y_T)$ is a sequence of generated tokens, $y_t \sim p_\theta(y_t | y_1, \ldots, y_{t-1})$.

In practice, the expected gradient can be approximated using a single Monte-Carlo sample from $p_\theta$. Using the sum of our target metrics as a reward function brings us to the final expression for our REINFORCE loss component:

$$\mathcal{L}_R(\theta) = -(\text{EM}(y) + \text{ES}(y)) \sum_{t=1}^{T} \log p_\theta(y_t | y_1, \ldots, y_{t-1}), \tag{2}$$

where $ES(y)$ and $EM(y)$ are the EM and ES metrics computed from a model rollout with greedy approach. Greedy generation prevents us from using the variance-reducing baseline term from Rennie et al. (2017), which is necessary in standard REINFORCE due to stability issues. However, as discussed in Section 3.6, this limitation is offset by the additional components of our final loss function 4.

### 3.4 COSINE ALIGNMENT LOSS

While training the projection from encoder representations into the LLM embedding space in our initial experiments, we observed that the projection MLP often collapsed to an almost one-dimensional subspace: the angles between projected vectors converged to nearly zero across most pairs (see Figure 2b), while the encoder itself is expressive, producing embeddings with pairwise cosine similarities broadly distributed in the range $[0.0, 1.0]$ (see Figure 2a).

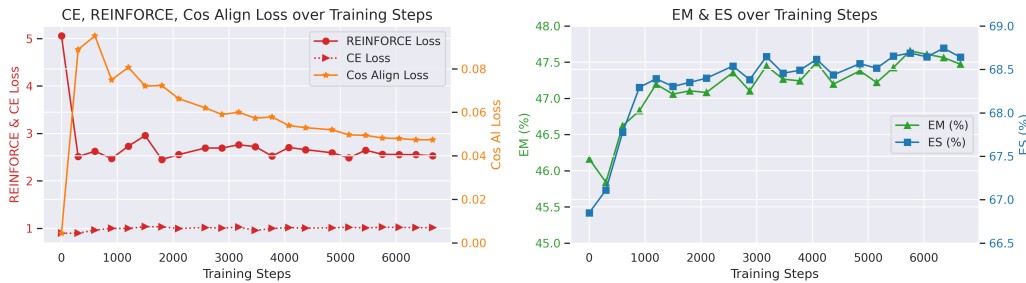

Figure 3: Relationship between the three loss components (Cross-Entropy, REINFORCE, and Cosine Alignment) and the evaluation metrics Exact Match (EM) and Edit Similarity (ES).

This behavior is undesirable, since we aim to preserve the distinctions between retrieved text chunks. To address this collapse and retain the relative differences among encoder embeddings after projection, we introduce a specialized *Cosine Alignment Loss*:

$$\mathcal{L}_A(\theta) = \frac{1}{\sqrt{2}} \|S_C(y_{\text{enc}}) - S_C(y_{\text{proj}})\|_F, \tag{3}$$

where $S_C$ denotes the cosine similarity matrix between vectors of the output batch, and $y_{\text{enc}}$ and $y_{\text{proj}}$ represent the encoder and projection output batches, respectively. This loss enforces preservation of pairwise cosine similarities within a batch by minimizing the mean squared error (MSE) between the similarity matrices. The factor $\frac{1}{\sqrt{2}}$ compensates for the symmetry of the cosine similarity matrix.

The loss formulation in Equation 3 helps preserve relative differences between retrieved contexts. Figure 2c shows the resulting cosine distance matrix for 100 random samples, demonstrating that the projections remain mostly well-separated and their cosine distance matrix has the same structure as the original embeddings' matrix. In contrast to that, collapsed projector outputs form a single indistinguishable representation, as seen in Figure 2b.

### 3.5 KL-LOSS

In contrast to the findings reported by Cheng et al. (2024), we observe that the KL-divergence loss between models trained with compressed versus uncompressed retrieved context does not improve prediction quality. This outcome directly conflicts with the results presented in the xRAG paper, where the KL-loss term was assigned a weight of 2.0 relative to the NLL-loss weight of 1.0. Their choice was based on ablation studies with instruction-tuned models on QA datasets, where they attributed performance gains primarily to the KL-loss component, arguing that it improved the model's resilience rate—defined as the proportion of cases in which responses remained correct both before and after retrieval augmentation.

To thoroughly evaluate the xRAG approach to projector training, we perform an ablation study using multiple loss formulations, including a full reproduction of the xRAG loss described above. As shown in Table 1, this loss configuration yields performance below the unaugmented model baseline.

### 3.6 FINAL LOSS FUNCTION

We optimize our model using the following composite loss function:

$$\mathcal{L}(\theta) = \alpha_{CE}\mathcal{L}_{CE}(\theta) + \alpha_R\mathcal{L}_R(\theta) + \alpha_A\mathcal{L}_A(\theta), \tag{4}$$

where the coefficients $\alpha_{CE}$, $\alpha_R$, and $\alpha_A$ are weighting factors. These weights are selected through hyperparameter tuning using the Optuna framework (Akiba et al., 2019)[3]. These and other training hyperparameters fot all trained models are listed in Appendix C. The loss dynamic and its correspondence to target metrics EM and ES can be seen in Figure 3.

---

[3]https://optuna.org

| Method | $\alpha_{CE}$ | $\alpha_R$ | $\alpha_A$ | $\alpha_{KL}$ | CE Loss ↓ | EM ↑ | ES ↑ |
|---|---|---|---|---|---|---|---|
| Base Model w/o CFC | - | - | - | - | 0.97 | 45.97 | 66.57 |
| Base Model w/ CFC | - | - | - | - | 0.99 | 50.87 | 69.43 |
| LLaVA (Liu et al., 2023) | 1.0 | 0.0 | 0.0 | 0.0 | **0.80** | 38.57 | 63.6 |
| REINFORCE-only | 0.0 | 1.0 | 0.0 | 0.0 | 5.18 | 40.61 | 63.91 |
| CE + Cos Align | 0.9 | 0.0 | 0.1 | 0.0 | 0.89 | 42.3 | 64.43 |
| xRAG (Cheng et al., 2024) | 1.0 | 0.0 | 0.0 | 2.0 | _0.84_ | _44.0_ | _65.5_ |
| **LlavaCode (ours)** | 0.9 | 0.1 | 0.1 | 0.0 | 1.02 | **47.66** | **68.74** |

Table 1: Ablation studies on different approaches to projection training. Result without context augmentation is denoted "w/o CFC". Result with uncompressed cross-file context is denoted by "w/ CFC" and highlighted with gold. $\alpha_{KL}$ denotes the weight of KL-loss. Other loss components are as in Section 3.6. Metrics are reported on evaluation subset of our dataset ($\approx 4.3$k samples).

As noted in Section 3.3, we use standard REINFORCE without a variance-reduction baseline, an approach reported to be weaker then SCST, as reported by Rennie et al. (2017). However, the CE loss term serves as a strong stabilizer, reducing the need for such a baseline in the REINFORCE component. As shown in Table 1, REINFORCE alone diverges, whereas in the presence of CE loss REINFORCE trains successfully—confirming the stability issues of standard REINFORCE when used in isolation.

## 4 EXPERIMENTS

### 4.1 DATASET

We trained our models on the Python subset of The Stack dataset (Kocetkov et al., 2022). To ensure dataset quality, we organized files by repository and applied the following filtering steps: we excluded repositories with fewer than 50 stars, fewer than 5 files, or files containing fewer than 3 import statements. After filtering, the dataset contained approximately 150k code completion samples, each paired with at least ten relevant cross-file context snippets. Relevant examples were identified using the Jaccard text similarity metric applied to code chunks drawn from the surrounding repository (excluding the current file used for code completion).

The code completion task takes fill-in-the-middle (FIM) format, where the left and right contexts are provided and the missing middle segment must be generated by the LLM. Each target segment consists of $n_t$ lines ($1 \leq n_t \leq 9$), with $n_t$ sampled from a Poisson distribution. Code was segmented into chunks of $10 \times n_t$ lines with an overlap of $5 \times n_t$ lines. We tried to enhance RAG with more sophisticated code search techniques, such as utilizing cosine distance between text embeddings from various models, but Jaccard showed the best results. For more information, see Appendix F.

During training, we use all available length of the target, while evaluation is performed specifically on single line completions. For evaluation, the dataset was split at the repository level to ensure that samples from a given repository appeared exclusively in either the training or validation set. Additionally, we remove all leading and trailing whitespace to ensure that ES metric is not artificially inflated.

### 4.2 TRAINING

We train 2- and 3-layer MLP projection modules that map sentence encoder outputs (e.g., UniX-Coder or Qwen3Embedding) into the dimension of code LLM embeddings. For each sample, the top 10 cross-file contexts, encoded and projected into LLM representations at the time of RAG database creation, are concatenated with the code completion prompt embeddings before being passed to the LLM. When comparing against the LLM with non-compressed text context, the top 10 retrieved contexts are concatenated into one sequence, truncated to 512 tokens and then concatenated with the same code completion prompt. The code completion prompt budget (without retrieved context) is 2k tokens for both methods. As a result, the input sequence length in our approach is 502 tokens shorter than in conventional RAG.

| Benchmark | Model | Seq Length | EM ↑ | ES ↑ | CodeBLEU↑ |
|---|---|---|---|---|---|
| | w/o CFC | 2000 | 45.64 | 71.75 | 55.82 |
| CCEvalLong (Wu et al., 2024) | w/ CFC | 2500 | **49.74** | 73.2 | **58.01** |
| | LlavaCode (ours) | 2010 | 47.16 | **73.35** | 57.46 |
| | w/o CFC | 2000 | 58.13 | 77.98 | 58.09 |
| | w/ CFC | 2500 | **64.56** | **81.69** | **64.34** |
| RepoEval (Zhang et al., 2023) | LlavaCode (ours) | 2010 | 60.56 | 80.46 | 59.67 |
| | Context Pruning | 2010 | 57.91 | 78.1 | 58.05 |
| | Context Summarization | 2010 | 58.04 | 78.26 | 58.23 |
| | CodePromptZip (He et al., 2025) | 2010 | 57.91 | 78.3 | 58.26 |
| | w/o CFC | 2000 | 48.75 | 74.13 | 52.32 |
| RepoEval Api (Zhang et al., 2023) | w/ CFC | 2500 | **55** | **79.31** | **57.6** |
| | LlavaCode(ours) | 2010 | 51.31 | 77.95 | 55.56 |

Table 2: Results on code completion benchmarks. Qwen2.5Coder-7B was used as the base code-generating LLM for all methods.

During training, only the projection weights are updated, while both the encoder and the LLM remain frozen. Optimization is performed using the joint loss described in Section 3.6, which combines all three loss components. Cross-Entropy is only computed over the sequence after the `<|fim_middle|>` special token. For REINFORCE loss, we generate 50 tokens using greedy decoding and evaluate EM and ES metrics on the obtained sequence. A full list of training hyperparameters, including the coefficients for each loss component, is provided in Appendix C.

Table 1 presents the results of our ablation studies across different loss formulations, comparing four configurations: Cross-Entropy only (LLaVA-style), REINFORCE only, Cross-Entropy with Cosine Alignment, and Cross-Entropy with KL Loss (the xRAG objective). As discussed in Section 3.2, relying solely on the Cross-Entropy objective degrades performance on both EM and ES metrics. Conversely, optimizing exclusively with the REINFORCE loss leads to uncontrolled entropy growth and fails to outperform the w/o CFC baseline, due to the absence of variance-reducing baseline. In contrast, only a carefully balanced combination of all three loss components (Section 3.6) yields consistent improvements across the target metrics (Figure 3).

The ablation in Table 3 studies the effect of encoder choice and projection depth. We evaluate two encoders and code modalities: UniXCoder with AST representations of retrieved code, and the Qwen-3-Embedding-0.6B model with retrieved code. Qwen-3-Embedding-0.6B used as the retrieved-context compressor outperforms UniXCoder. A three-layer MLP projection further improves both EM and ES but increases the number of trainable parameters by roughly 4×.

Our main results on several well-known code-completion benchmarks are presented in Table 2. We compare our approach—retrieved-context compression via LlavaCode—against a base model with no additional context, as well as a model that uses uncompressed retrieved context. For the RepoEval benchmark, we further compare our method with other context-compression techniques that achieve a similar compression ratio, such as token pruning and summarization. Pruning and summarization were performed using the Qwen2.5Coder-7B Instruct model. Additionally, we compare our approach with CodePromptZip (He et al., 2025), a code-focused technique designed to reduce context length. As shown in the table, the level of extreme compression achieved by LlavaCode leads to severely bad performance for methods that operate in token space, which typically perform best only at moderate compression ratios (around 0.3).

Despite negligible latency impact introduced by the additional 10 tokens, our approach surpasses the no-CFC baseline on EM and ES metrics by a sizable margin, which makes our approach preferable in latency-limited environments, such as IDE code completion, where vanilla RAG introduces noticeable latency impact in the range of 20-38%. Detailed latency measurements are presented in Section 5 and in Tables 8, 9. When compared to other context compression methods, LlavaCode significantly outperforms them; the alternative approaches show virtually no improvement over the base model's prediction quality at such extreme compression rates.

Concrete code completion examples, along with resilience rates from the benchmark results, are presented in Table 6 and Figure 4 in Appendix B. Table 5 shows the results of running the original xRAG pipeline with openly available model[4] on RepoEval benchmark.

---

[4]https://huggingface.co/Hannibal046/xrag-7b

| Encoder | Modality | Projection | # Trainable Parameters | EM | ES |
|---|---|---|---|---|---|
| UniXCoder | AST | 2-layer MLP | 3.5M | 46.69 | 67.65 |
| UniXCoder | AST | 3-layer MLP | 16.5M | 46.94 | 68.26 |
| Qwen3Embedding | Code | 2-layer MLP | 3.9M | 47.01 | 68.15 |
| Qwen3Embedding | Code | 3-layer MLP | 17.3M | **47.66** | **68.74** |

Table 3: Comparison of different encoders and projection heads with their trainable parameters and performance metrics. Differences in the number of trainable parameters emerge from the encoder output dimension and the number of MLP layers. All configurations were trained for $\approx 6{,}600$ training steps (3 epochs).

## 5 SPEEDUP ESTIMATION

In our LlavaCode pipeline, the Encoder + Projector processes contextual chunks from the surrounding repository during the RAG database build—typically when the IDE indexes the project. As a result, at inference time the system simply appends the precomputed projections to the code-completion prompt. This means that the primary factor influencing how quickly the user receives a completion suggestion is the sequence length. In this section we demonstrate the practical benefits of sequence length reduction, provided by LlavaCode.

Two deployment patterns dominate today's LLM serving landscape. First, *prefill–decode mixing*, uses single engine which interleaves chunks from prompt prefill with decoding passes across requests. For instance, one of the inference engines, which utilizes this approach, is vLLM framework Kwon et al. (2023). Second, *disaggregated prefill-decode*, when prefill and decode run on separate GPU pools or nodes (possibly on different clusters) with independent resource plans. An example of an engine that uses this approach is DistServe Zhong et al. (2024).

Colocating prefill and decode is utilization-friendly and achieves high throughput on single machines via memory-efficient KV management and continuous batching. However, prefill and decode contend for distinct resources and interfere with each other, which makes it hard to independently control TTFT (*time to first token*) and TPOT (*time per output token*) under enterprise's Service Level Agreement (SLA). As a result, systems are often over-provisioned with hardware to satisfy both metrics. Agrawal et al. (2024); Wang et al. (2024)

Separating the phases decouples resource allocation and parallelism strategies, eliminating prefill–decode interference and enabling direct tuning of TTFT (prefill stage) and TPOT (decode stage). Operationally, it simplifies capacity planning and horizontal scaling because each fleet can scale along its own bottleneck. User will operate over IDE in interactive manner, so TTFT of code completion LLM is the main metric to which the experience is sensitive, since, as soon as tokens start generating, user can start reviewing code suggestions.

For disaggregated serving (transformers) and colocated prefill–decode (vllm) the results are shown in Table 8. For performance measurements, we report scaling metrics for both inference patterns. For benchmarking, we implement separate prefill and decode workers using the transformers runtime Wolf et al. (2020). More detailed results, including TPOT metric, are listed in Appendix D.

Reducing prompt length primarily improves TTFT; in colocated engines it often yields limited gains on decode-side TPOT, which remains dominated by iterative decode dynamics and batching. Under disaggregation, the effect becomes more predictable: shorter contexts directly reduce prefill latency and lower the number of GPUs requirements to handle the same load while leaving decode behavior isolated, allowing clearer SLA tuning for each phase.

## 6 CONCLUSIONS AND FUTURE WORK

In conclusion, we propose a novel pipeline for retrieval augmented code generation using LLaVA-like projection of retrieved code chunks into LLM embeddings, which significantly increases the quality of code completions, while introducing negligible effect on latency. Compared to full RAG, our approach results in 20-38% better prompt processing speed and latency metrics, which is criti-

| Sequence compression | Model | TTFT transformers | TTFT vllm |
|---|---|---|---|
| $2500 \rightarrow 2010 \downarrow 20\%$ | Qwen2.5-Coder-1.5B | $198.2 \rightarrow 156.6 \downarrow 21\%$ | $74.7 \rightarrow 68.2 \downarrow 9\%$ |
| | Qwen2.5-Coder-7B | $668.6 \rightarrow 541.1 \downarrow 19\%$ | $198.3 \rightarrow 166.5 \downarrow 16\%$ |
| | Qwen2.5-Coder-14B | $822.8 \rightarrow 661.3 \downarrow 20\%$ | $349.8 \rightarrow 291.7 \downarrow 17\%$ |
| $2000 \rightarrow 1510 \downarrow 24\%$ | Qwen2.5-Coder-1.5B | $157.4 \rightarrow 113.4 \downarrow 28\%$ | $65.3 \rightarrow 58.4 \downarrow 11\%$ |
| | Qwen2.5-Coder-7B | $540.0 \rightarrow 406.8 \downarrow 25\%$ | $179.0 \rightarrow 134.0 \downarrow 25\%$ |
| | Qwen2.5-Coder-14B | $662.2 \rightarrow 496.3 \downarrow 25\%$ | $291.5 \rightarrow 232.9 \downarrow 20\%$ |
| $1500 \rightarrow 1010 \downarrow 33\%$ | Qwen2.5-Coder-1.5B | $112.2 \rightarrow 69.7 \downarrow 38\%$ | $58.9 \rightarrow 50.3 \downarrow 15\%$ |
| | Qwen2.5-Coder-7B | $406.4 \rightarrow 282.2 \downarrow 31\%$ | $138.0 \rightarrow 112.4 \downarrow 19\%$ |
| | Qwen2.5-Coder-14B | $495.6 \rightarrow 339.6 \downarrow 31\%$ | $238.2 \rightarrow 174.6 \downarrow 27\%$ |

Table 4: For disaggregated inference deployment (measured with transformers library) context compression directly leads to almost same decrease of TTFT. This way, response for user's query start generating and showing to user much earlier. For prefill-decode mixing, as described in Section 5, speedup is lower than context compression, due to decode workload dominating on latency. Measured on NVIDIA A100.

cal for code completion applications, while maintaining slightly worse, but comparable generation quality.

To the best of our knowledge, our work is the first among the LLaVA-like approaches to apply compression to code generation models, explore the addition of semantically rich code modalities, utilize base models instead of instruction-tuned models, and apply reinforcement learning to train the projection for downstream code-completion tasks. We achieve this by training only a lightweight projection module, without modifying the embedding model and code generation LLM.

Using the REINFORCE algorithm, we directly optimize ES and EM metrics, which are closely linked to positive user experience in interactive code completion environments. Additionally, by introducing a novel Cosine Alignment Loss, we preserve document-level distinctions after projection. Moreover, we demonstrate that all previously proposed methods for training a projector fail on code completion tasks, and design a more sophisticated loss function that consistently improves the target metrics.

Future work could investigate state-of-the-art RL methods to improve alignment with EM/ES metrics, such as PPO or GRPO. In addition, as new encoders for graph modalities are developed, our approach could be re-evaluated using these improved architectures. Finally, our current experiments are limited to the Python subset of The Stack dataset; extending the evaluation to other widely used languages such as Java, C#, and beyond would provide a broader assessment of the method's generality.

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

| | Sequence Length | EM ↑ | ES ↑ | CodeBLEU↑ |
|---|---|---|---|---|
| w/ CFC | 2000 | 0.19 | 14.05 | 3.88 |
| xRAG | 2010 | 0.19 | 13.42 | 4.10 |

Table 5: Results of openly available xRAG model[5] on RepoEval benchmark.

An Yang, Anfeng Li, Baosong Yang, Beichen Zhang, Binyuan Hui, Bo Zheng, Bowen Yu, Chang Gao, Chengen Huang, Chenxu Lv, Chujie Zheng, Dayiheng Liu, Fan Zhou, Fei Huang, Feng Hu, Hao Ge, Haoran Wei, Huan Lin, Jialong Tang, Jian Yang, Jianhong Tu, Jianwei Zhang, Jianxin Yang, Jiaxi Yang, Jing Zhou, Jingren Zhou, Junyang Lin, Kai Dang, Keqin Bao, Kexin Yang, Le Yu, Lianghao Deng, Mei Li, Mingfeng Xue, Mingze Li, Pei Zhang, Peng Wang, Qin Zhu, Rui Men, Ruize Gao, Shixuan Liu, Shuang Luo, Tianhao Li, Tianyi Tang, Wenbiao Yin, Xingzhang Ren, Xinyu Wang, Xinyu Zhang, Xuancheng Ren, Yang Fan, Yang Su, Yichang Zhang, Yinger Zhang, Yu Wan, Yuqiong Liu, Zekun Wang, Zeyu Cui, Zhenru Zhang, Zhipeng Zhou, and Zihan Qiu. Qwen3 technical report, 2025. URL https://arxiv.org/abs/2505.09388.

Tatiana Zemskova and Dmitry Yudin. 3dgraphllm: Combining semantic graphs and large language models for 3d scene understanding, 2025. URL https://arxiv.org/abs/2412.18450.

Fengji Zhang, Bei Chen, Yue Zhang, Jacky Keung, Jin Liu, Daoguang Zan, Yi Mao, Jian-Guang Lou, and Weizhu Chen. RepoCoder: Repository-level code completion through iterative retrieval and generation. In Houda Bouamor, Juan Pino, and Kalika Bali (eds.), *Proceedings of the 2023 Conference on Empirical Methods in Natural Language Processing*, pp. 2471–2484, Singapore, December 2023. Association for Computational Linguistics. doi: 10.18653/v1/2023.emnlp-main. 151. URL https://aclanthology.org/2023.emnlp-main.151/.

Yanzhao Zhang, Mingxin Li, Dingkun Long, Xin Zhang, Huan Lin, Baosong Yang, Pengjun Xie, An Yang, Dayiheng Liu, Junyang Lin, Fei Huang, and Jingren Zhou. Qwen3 embedding: Advancing text embedding and reranking through foundation models, 2025. URL https://arxiv.org/abs/2506.05176.

Yinmin Zhong, Shengyu Liu, Junda Chen, Jianbo Hu, Yibo Zhu, Xuanzhe Liu, Xin Jin, and Hao Zhang. {DistServe}: Disaggregating prefill and decoding for goodput-optimized large language model serving. In *18th USENIX Symposium on Operating Systems Design and Implementation (OSDI 24)*, pp. 193–210, 2024.

## A   LLM USAGE STATEMENT

We used ChatGPT-5 and ChatGPT-4o to correct grammatical and stylistic errors, condense text, perform translations, and rephrase content.

## B   DETAILED RESULTS

Table 6 presents examples illustrating how LlavaCode handled code completion tasks from our benchmarks, compared with both the baseline model without cross-file context (w/o CFC) and the model using uncompressed cross-file context (w/ CFC).

Figure 4 shows the resilience rates—the percentage of instances where the model's response remains correct both before and after retrieval augmentation—for LlavaCode compared to a model using uncompressed cross-file context. Overall, the rates are comparable, with LlavaCode showing a slight advantage on average.

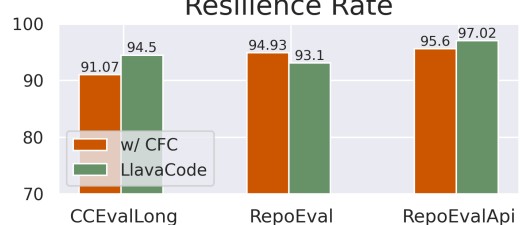

Figure 4: Resilience rate of two augmentation methods: uncompressed retrieved cross-file context (w/o CFC) and LlavaCode over a Qwen2.5Coder-7B baseline without retrieval augmentation.

Table 5 presents the results of running the original xRAG pipeline on RepoEval. Since xRAG relies on the general-purpose, non code-specific Mistral-7B model and its compression mechanism was trained on retrieved documents for downstream QA tasks, we do not include these results in our main comparison in Table 2. Nevertheless, the results highlight that context compression techniques designed for natural language documents are insufficient for code, leading to either a marginal increase in CodeBLEU or a decrease in other metrics such as Exact Similarity.

| Code Completion | |
|---|---|
| groundtruth | `pos += slice_length` |
| w/o CFC | `return` |
| w/ CFC | `yield ""` |
| LlavaCode | `pos += slice_length` |
| groundtruth | `fields_values[name] = NAO` |
| w/o CFC | `if name in fields:` |
| w/ CFC | `if name in fields:` |
| LlavaCode | `fields_values[name] = NAO` |
| groundtruth | `trial_metadata: Iterable[UnitMetadataUpdate],` |
| w/o CFC | `trial_metadata: Iterable[` |
| w/ CFC | `trial_metadata: Iterable[UnitMetadataUpdate],` |
| LlavaCode | `trial_metadata: Iterable[key_value_pb2.KeyValue],` |
| groundtruth | `from jax import random` |
| w/o CFC | `import random` |
| w/ CFC | `from jax import random` |
| LlavaCode | `from jax import random` |
| groundtruth | `from fortuna.prob_model.posterior.map.map_stat` |
| w/o CFC | `cfg = compile_config(cfg, create_cfg=create_cfg)` |
| w/ CFC | `from fortuna.prob_model.posterior.map.map_ste` |
| LlavaCode | `# Get env_fn from env_setting.` |

Table 6: Examples of generated code for the code-completion benchmarks reported in Table 2. "w/o CFC" denotes model without cross-file context, and "w/ CFC" denotes model using uncompressed cross-file context. Although some benchmark tasks involve multi-line completions, only one-line examples are shown here to fit more examples.

## C  Training Parameters

For our primary evaluations, we used the Qwen2.5Coder family of models, with the Qwen-3-Embedding-0.6B model serving as the encoder. A three-layer MLP was employed as the projector, mapping from the encoder dimension to twice the embedding size of the LLM, and finally down to the LLM's embedding size. A GELU activation and a LayerNorm were applied between the first and second layers, and again between the second and final layer. Training hyperparameters for different model sizes are described in 7.

| Hyperparameter | Qwen2.5Coder-1.5B | Qwen2.5Coder-7B |
|---|---|---|
| optimizer | AdamW | AdamW |
| alpha Cosine Alignment | 0.1 | 0.2 |
| alpha Cross-Entropy | 0.9 | 0.9 |
| alpha REINFORCE | 0.1 | 0.05 |
| learning rate | 1e-3 | 1e-4 |
| lr scheduler type | cosine | cosine |
| warmup ratio | 0.03 | 0.04 |
| weight decay | 0.0 | 0.0 |
| epochs | 3 | 3 |
| effective batch size | 66 | 64 |
| train samples | 150k | 150k |

Table 7: Hyperparameters for projection training

| Sequence compression | Model | TTFT | TPOT |
|---|---|---|---|
| 2500→ 2010↓ 20% | Qwen2.5-Coder-1.5B | 198.2 → 156.6↓ 21% | 23.6 → 23.2↓ 2% |
| | Qwen2.5-Coder-7B | 668.6 → 541.1↓ 19% | 27.2 → 25.4↓ 7% |
| | Qwen2.5-Coder-14B | 822.8 → 661.3↓ 20% | 58.5 → 52.8↓ 10% |
| 2000→ 1510↓ 24% | Qwen2.5-Coder-1.5B | 157.4 → 113.4↓ 28% | 23.5 → 23.1↓ 2% |
| | Qwen2.5-Coder-7B | 540.0 → 406.8↓ 25% | 25.1 → 24.1↓ 4% |
| | Qwen2.5-Coder-14B | 662.2 → 496.3↓ 25% | 52.7 → 47.0↓ 11% |
| 1500→ 1010↓ 33% | Qwen2.5-Coder-1.5B | 112.2 → 69.7↓ 38% | 23.2 → 23.5↑ 1% |
| | Qwen2.5-Coder-7B | 406.4 → 282.2↓ 31% | 24.1 → 24.2 |
| | Qwen2.5-Coder-14B | 495.6 → 339.6↓ 31% | 46.8 → 41.1↓ 12% |

Table 8: For disaggregated inference deployment (measured with transformers library) context compression directly leads to almost same decrease of TTFT. This way, response for user's query start generating and showing to user much earlier. Measured on a single NVIDIA A100.

| Sequence compression | Model | TTFT | TPOT |
|---|---|---|---|
| 2500→ 2010↓ 20% | Qwen2.5-Coder-1.5B | 74.7 → 68.2↓ 9% | 5.3 → 5.3 |
| | Qwen2.5-Coder-7B | 198.3 → 166.5↓ 16% | 11.7 → 11.7 |
| | Qwen2.5-Coder-14B | 349.8 → 291.7↓ 17% | 22.3 → 21.8↓ 2% |
| 2000→ 1510↓ 24% | Qwen2.5-Coder-1.5B | 65.3 → 58.4↓ 11% | 5.3 → 5.6↑ 5% |
| | Qwen2.5-Coder-7B | 179.0 → 134.0↓ 25% | 11.7 → 11.6↓ 1% |
| | Qwen2.5-Coder-14B | 291.5 → 232.9↓ 20% | 21.8 → 21.8 |
| 1500→ 1010↓ 33% | Qwen2.5-Coder-1.5B | 58.9 → 50.3↓ 15% | 5.4 → 6.3↑ 16% |
| | Qwen2.5-Coder-7B | 138.0 → 112.4↓ 19% | 11.6 → 11.5↓ 1% |
| | Qwen2.5-Coder-14B | 238.2 → 174.6↓ 27% | 21.7 → 21.4↓ 1% |

Table 9: For prefill-decode mixing, context compression leads to more efficiency. But, as described in Section 5, speedup is lower than for context compression, due to decode workload dominating on latency. Measured on NVIDIA A100.

## D  DETAILED LATENCY AND LOAD MEASUREMENTS

This section expands on the results presented in Section 5, including TTOP measurements as shown in Tables 8 and 9, as well as latency reduction measurements for prefill-only regime (1-token generation), reported in Tables 10 and 11.

## E  ON PRETRAINING OF THE PROJECTION MODULE

Whereas most prior work adopts two-stage training, we use a single-stage pipeline based on a composite loss function, discussed in Section 3.6. For completeness, we also evaluated a conventional two-stage pretrain–finetune pipeline for projection training.

| Sequence compression | Model | TTFT |
|---|---|---|
| 2500→ 2010↓ 20% | Qwen2.5-Coder-1.5B | 198.2 → 159.3↓ 20% |
| 2500→ 2010↓ 20% | Qwen2.5-Coder-7B | 668.1 → 539.1↓ 19% |
| 2500→ 2010↓ 20% | Qwen2.5-Coder-14B | 820.9 → 661.1↓ 19% |
| 2000→ 1510↓ 24% | Qwen2.5-Coder-1.5B | 159.9 → 121.2↓ 24% |
| 2000→ 1510↓ 24% | Qwen2.5-Coder-7B | 539.1 → 406.3↓ 25% |
| 2000→ 1510↓ 24% | Qwen2.5-Coder-14B | 660.6 → 495.8↓ 25% |
| 1500→ 1010↓ 33% | Qwen2.5-Coder-1.5B | 120.7 → 75.5↓ 37% |
| 1500→ 1010↓ 33% | Qwen2.5-Coder-7B | 405.4 → 281.0↓ 31% |
| 1500→ 1010↓ 33% | Qwen2.5-Coder-14B | 494.9 → 339.6↓ 31% |

Table 10: Latency reduction in prefill-only regime (generation of 1 token). Transformers library.

| Sequence compression | Model | TTFT |
|---|---|---|
| 2500→ 2010↓ 20% | Qwen2.5-Coder-7B | 197.4 → 165.9↓ 16% |
| 2500→ 2010↓ 20% | Qwen2.5-Coder-14B | 351.5 → 291.2↓ 17% |
| 2500→ 2010↓ 20% | Qwen2.5-Coder-1.5B | 79.1 → 67.0↓ 15% |
| 2000→ 1510↓ 24% | Qwen2.5-Coder-7B | 164.4 → 135.8↓ 17% |
| 2000→ 1510↓ 24% | Qwen2.5-Coder-14B | 290.0 → 240.9↓ 17% |
| 2000→ 1510↓ 24% | Qwen2.5-Coder-1.5B | 65.9 → 56.4↓ 14% |
| 1500→ 1010↓ 33% | Qwen2.5-Coder-7B | 136.5 → 104.6↓ 23% |
| 1500→ 1010↓ 33% | Qwen2.5-Coder-14B | 240.2 → 191.4↓ 20% |
| 1500→ 1010↓ 33% | Qwen2.5-Coder-1.5B | 56.4 → 48.7↓ 14% |

Table 11: Latency reduction in prefill-only regime (generation of 1 token). vLLM framework.

In prior work, pretraining often relies on parallel datasets, such as paraphrase pairs in xRAG or image–caption pairs in LLaVA. Inspired by xRAG, we experimented with a similar pretraining approach, attempting to reconstruct retrieved context chunks from projected vectors by optimizing the entropy loss. This approach did not yield improvements in the second stage of training, likely due to the entropy issues discussed in Section 3.2.

Kuratov et al. (2025) demonstrate that up to 1,568 tokens can be compressed into a single continuous "memory" token by treating the token as a trainable parameter and optimizing it via backpropagation with a cross-entropy reconstruction loss. Because these continuous tokens reconstruct to reference texts, we treat them as ground truth for training our projection layer. Concretely, we encode text with our encoder, project the resulting embeddings into a single token, and optimize a mixture of Mean Squared Error (MSE) and cosine-similarity (CS) losses between the projected embedding and the trained ground-truth compressed token.

However, the space spanned by the memory tokens proved to be highly non-smooth. For instance, identical text inputs could be compressed into vectors that are widely separated, and introducing even small perturbations to a learned memory token often results in reconstruction of completely different text. This leads to poor generalization for an MLP module attempting to map into this space. Consequently, learning a projection into such a space requires extreme overparameterization, effectively amounting to memorizing the entire dataset. As a result, we could only overfit on a small subset of memory tokens and were unable to learn a meaningful translation into the memory token space.

We leave the more sophisticated pretraining of the projection module for code compression to future work.

## F ON DIFFERENT RETRIEVAL TECHNIQUES

We evaluated multiple retrieval metrics for selecting the top-10 most relevant code chunks. Specifically, we compared sparse retrievers such as BM25 and Jaccard with dense retrievers based on cosine similarity over embeddings from UniXCoder and Jina v2. Each retriever-augmented model was benchmarked against a baseline model without any additional retrieved context. The comparison was conducted on a subset of 1,600 code completion tasks from our dataset as described in Section 4.1. The results show that Jaccard and UniXCoder achieved the best performance. Given its lower latency, we adopt Jaccard as the primary retrieval method in Section 4.1.

| Method | EM | ES |
|---|---|---|
| No CFC | 50.50 | 73.12 |
| BM25 | 55.56 | 76.5 |
| Jaccard | **56.19** | 76.68 |
| UniXCoder | 56.00 | **76.84** |
| Jina v2 | 54.31 | 75.56 |

Table 12: Comparison of different retrieval strategies.

