# OpenReview forum: "LlavaCode: Compressed Code Representations for Retrieval-Augmented Code Generation"
_ICLR.cc/2026/Conference — ICLR 2026 Conference Withdrawn Submission_

### Official Review · Reviewer_ogA2 · 2025-10-17

**Soundness:** 1
**Presentation:** 3
**Contribution:** 1
**Rating:** 2
**Confidence:** 5

**Summary:**

The paper proposes LlavaCode, a retrieval-augmented code generation framework that compresses retrieved code into dense embeddings aligned with the LLM’s input space. A lightweight projector is trained with a mix of cross-entropy, REINFORCE, and cosine-alignment losses to preserve semantic fidelity while reducing input length. Experiments on The Stack (Python) show improved EM and Edit Similarity with notably lower latency, suggesting LlavaCode’s potential for efficient IDE code completion.

**Strengths:**

The paper is easy to follow and well-structured. The motivation—to reduce latency and input length in retrieval-augmented code generation—is clearly articulated.

**Weaknesses:**

This paper has several serious weaknesses.

First, the technical contribution is limited. The idea of embedding or compressing retrieved contexts has been well explored in recent RAG literature, and this work does not offer meaningful methodological innovation beyond applying it to repository-level code generation.

Second, the experimental setup is unsound and fails to validate the proposed method’s effectiveness. The method is designed to compress retrieved context, yet in Section 4.1 the evaluation is performed under a single-file setting without any cross-file context, making the comparison with baselines largely meaningless. The paper provides no evidence that the proposed compression preserves model performance (e.g., EM, ES) when cross-file context is included. Under such a flawed setup, the reported efficiency gains have little discussion value.

Finally, the experimental scope is very limited. The authors do not evaluate on any standard RAG or code completion benchmarks, nor do they compare with strong existing baselines. All results are based on a small self-collected dataset and a single model, which is insufficient to support the claimed contributions.

**Questions:**

In Table 4.1, it seems that the main experiments are conducted under the in-file setting, without including cross-file context. Could you please clarify whether any of your evaluations are performed with cross-file context (CFC)? If not, how can we be confident that the proposed compression method does not compromise model performance (e.g., EM, ES) when cross-file context is provided?

---

> ### Author Response · Authors · 2025-11-21
>
> Thank you for your review. We are sorry to see a score of 2, and we appreciate the chance to clarify the concerns you raised.
>
> - Novelty:
>
> Recent compression approaches such as RepoFUSE and CodePromptZip operate in token space, which does not yield latency-free inference. LlavaCode instead compresses context in the embedding space, enabling substantially higher compression ratios. Meanwhile, prior projector-based methods that operate in the embedding domain (e.g., LLaVA, xRAG) have not explored code as a modality.
> Our primary contribution is showing that existing projector-training strategies (CE-only, CE+KL) fail on code and code-completion tasks, and proposing a new loss formulation that succeeds in this setting. Our loss combines: CE, an RL-based term not previously used for projector or compression training, and a novel cosine alignment loss.
> To highlight this, Table 1 now includes ablations comparing LLaVA-style (CE-only) and xRAG-style (CE+KL) training, with LlavaCode outperforming both.
>
> - Cross-file context:
>
> Could you please clarify what parts of the paper gave the impression that we evaluated only in the in-file setting, so that we could fix them? Subsection 4.2 states that we compress the cross-file retrieved context using our encoder–projector pipeline, concatenate it with a 2000-token completion prompt, and compare it with (i) the uncompressed cross-file context (w/ CFC) and (ii) no cross-file context (w/o CFC). This results and their sequence lengths shown in Table 2: 2010 (LlavaCode and some other compression techniques), 2512 (w/ CFC), and 2000 (w/o CFC). Thus, both LlavaCode and the w/ CFC baseline use cross-file context in all experiments.
>
> - Scope of evaluation:
>
> We agree that thorough comparisons on well-known benchmarks are essential. While Table 1 focuses on ablation studies, we have added results on CrossCodeEval, RepoEval, and RepoEvalAPI in Table 2.
> To broaden the scope, we have included additional metrics such as CodeBLEU, along with experiments on larger models (see Table 2). CodeBLUE incorporates SyntaxMatchScore and DataflowMatchScore and thus can be more suitable for code.
> Additionally, we include comparison to CodePromptZip and some other context compression techniques  in our results on RepoEval benchmark (Table 2) and show that LlavaCode significantly outperformed them under the 10-token context restriction.

---

### Official Review · Reviewer_Jpa6 · 2025-10-26

**Soundness:** 3
**Presentation:** 2
**Contribution:** 2
**Rating:** 4
**Confidence:** 3

**Summary:**

The paper presents LlavaCode, a method to compress the retrieved context into tokens by a separate frozen encoder and a learned projector, in order to reduce the TFFT latency. To train the projector, a composite loss combines token-level cross-entropy, a REINFORCE-style sequence reward on EM/ES, and a cosine-similarity alignment term. Results on a Python dataset show that the method can improve EM & ES over a non-RAG baseline (though quality remains below uncompressed RAG); and lower TFFT.

**Strengths:**

1. Clear and important problem framing
2. Its focus on TFFT latency reduction is relevant for interactive IDE settings
3. Empirical TTFT measurements under two serving regimes and across multiple model sizes provide a useful operational perspective.

**Weaknesses:**

1. Novelty claims can be overstated. Despite the difference in methodology, like the cosine-alignment loss and RL on new metrics, the evaluation lacks a direct comparison to xRAG as the baseline, which already demonstrated “one-token” RAG with frozen retriever and LLM. A discussion of prior code-focused prompt/context compression works, such as RepoFUSE and CodePromptZip, is also missing.
2. Section 3.1 states top-10 retrieved chunks per completion and 512-token truncation per chunk, which would imply thousands of extra tokens, yet the “w/ CFC” baseline adds only 512 tokens (2000→2512). To clarify, does the CFC baseline only include the top-1 retrieved chunk, or does it truncate the concatenated top-10 chunks to 512?
3.  The paper states the gradient is approximated “using a single Monte-Carlo sample from pθ,” but then computes rewards on greedy rollouts and omits a variance-reducing baseline (contrary to SCST). This produces a biased, high-variance estimator with unclear benefits, and there is no analysis of reward scaling or stability.
4. Experiments only use Qwen2.5-Coder-1.5B for quality evaluation; larger or different code LMs are not tested for accuracy (only TTFT). Also, the best EM/ES numbers still come from uncompressed cross-file context, indicating the compression method is not lossless -- an analysis on the quality–compression trade-off would be useful.
5. EM and ES are surface-form metrics that reward lexical overlap and exact phrasing; they can penalize semantically correct alternatives (e.g., different variable names or algorithmic implementations). Why not use execution-based metrics, or code-aware similarity metrics like CodeBLEU?
6. In section 5, when measuring the latency, does it take the encoder and projector inference latency into account? If not, the reported TTFT gains only reflect prefill compute savings under synthetic prompt shortening, not the actual compressed-RAG system.
7. Writing issues:
- typos like “freezed,” “weight dacay”
- wrong in-text citation format -- use \citep

**Questions:**

1. Consider adding stronger baselines like code-specific compression (RepoFUSE, CodePromptZip) and xRAG adapted to code.
2. Clarify why does “w/ CFC” add only 512 tokens?
3. Across different sizes of Qwen2.5-Coder, does projector weights transfer across backbones or require re-training?

---

> ### Author Response · Authors · 2025-11-21
>
> Thank you for your review. We are sorry to see a score of 4, and we appreciate the opportunity to clarify the questions and concerns you raised.
>
> - Lack of Comparison to xRAG:
>
> As noted in Section 3.2, xRAG trains its projector using CE + KL. In Section 3.5, we evaluate setting with the same loss function. To make the comparison clearer, we now include the CE + KL configuration (xRAG-style training) in our ablation studies (Table 1), and we additionally report results using the original publicly available xRAG model on code-completion tasks in Appendix B Table 5.
>
> - On 512-Token Truncation:
>
> We agree that the phrasing in Subsection 4.1 could be misinterpreted to mean that each of the top-10 retrieved contexts was truncated individually to 512 tokens. In fact, we concatenate the top-10 retrieved contexts in relevance order and then truncate the combined sequence to 512 tokens—following standard practice in retrieval-augmented generation pipelines (e.g., RepoFormer). We have updated the text in Subsection 4.1 to remove this ambiguity.
>
> - RepoFUSE and CodePromptZip:
>
> We include comparison to CodePromptZip in our results on benchmarks (Table 2) along with some other context compression techniques and show that LlavaCode significantly outperformed them under the 10-token context restriction.
> Regarding RepoFUSE, it is not directly comparable to LlavaCode because it operates as a retrieval strategy—selecting which cross-file contexts to include based on a token budget—rather than performing embedding-level compression. In order to match LlavaCode’s compression rate, RepoFUSE would need to slice the repository into chunks of fewer than 10 tokens, which is too small to contain meaningful semantic information. Therefore, a fair comparison of compression rates between RepoFUSE and LlavaCode is not feasible.
>
> - Variance-Reducing Baseline in REINFORCE:
>
> While a formal REINFORCE implementation often uses a baseline, our setting differs in key ways that mitigate the need for one: i)our task is code completion, where generation is typically greedy; we therefore use greedy decoding for the rollouts. ii) Exploration is handled through the CE term in the final loss, which also stabilizes training and prevents the model from drifting away from the ground-truth distribution. iii) Table 1 shows that REINFORCE without CE diverges, whereas CE + REINFORCE succeeds—demonstrating that CE provides the necessary stabilization. iv) Our rewards (EM and ES) are bounded in [0,1], preventing gradient explosion. We further scale the REINFORCE term by α=0.1.
> Thus, although our approach differs from classical REINFORCE with a baseline, deterministic rollouts, bounded rewards, and CE-based regularization collectively address variance and stability concerns for our setting. We have added these points to the Section 3.6.
>
> - Scope of Evaluation:
>
> To broaden the scope, we have included additional metrics such as CodeBLEU, along with experiments on larger models (see Table 2). CodeBLUE incorporates SyntaxMatchScore and DataflowMatchScore and thus can be more suitable for code.
>
> Considering the execution-based metrics: this is a tricky evaluation to perform, since code completion benchmarks often deal with uncompilable chunks of code (unlike function generation benchmarks), or consider repositories poorly covered by unit tests.
> We focused on EM and ES,  because these are the metrics that are very often reported in the literature on code completion. Additionally, EM can serve as a lower limit of the pass@1 execution metric.
>
> Nevertheless, we are in the process of compiling an appropriate benchmark to measure a pass@k metric and will try to add these results in a time before the deadline.
>
> - Quality–Compression Trade-off:
>
> Our goal was latency-free RACG, which is of high demand for modern IDEs, most of which now include the ai-assistant. We agree though, that a variation of LlavaCode architecture that performs milder compression is an interesting topic and we leave this for future research.
> Additionally, some information loss during compression may sometimes be advantageous for retrieval-augmented generation: irrelevant retrieved context can harm model predictions. The xRAG paper and RepoBench benchmark highlighted this effect. Our compression even has a slightly better resilience rate than the uncompressed context approach (Figure 4 in Appendix B).
>
> - Encoder and Projector Inference Latency:
>
> As noted at the end of Section 3.1, we retrieve precomputed projections from the RAG database. Encoder and projector inference therefore occurs during database construction (e.g., during IDE indexing), not during code-completion inference. We have reiterated this detail in Section 5 to clarify why our runtime latency remains minimal.
>
> - Projector Weight Transfer:
>
> Because the projector outputs vectors in the LLM’s embedding space, its compatibility depends on the embedding dimension. For the models, discussed in the paper, the embedding dimension differs.

---

> > ### Comment · Reviewer_Jpa6 · 2025-11-26
> >
> > Dear Authors,
> >
> > Thank you for your response and the additional experiments. I have a few brief follow-up questions. Please note that no further experiments are needed; a one-sentence clarification for each point will suffice:
> >
> > - Regarding your response to W1: In Table 5, why is the performance on the RepoEval benchmark so low? Is that because of the backbone model?
> >
> > - For Table 5, does "w/ CFC" refer to the same configuration as the "w/ CFC" entry in Table 2 (line 382)?
> >
> > - Could you discuss the comparison to xRAG regarding compression efficiency?
> >
> > - Regarding Table 1: The "w/ CFC" method still dominates on most metrics. Therefore, measuring the trade-off between quality and compression becomes important. Have you assessed GFLOPs, like in xRAG?

---

### Official Review · Reviewer_H5NU · 2025-10-28

**Soundness:** 2
**Presentation:** 2
**Contribution:** 2
**Rating:** 2
**Confidence:** 4

**Summary:**

This paper focuses on compressed representations of retrieved texts to handle the long context problem in RAG. Inspired by Llava, the authors use an embedding model and a projection layer to condense the retrieved text representations into one token representation that is fed to the LLM for generation. To address several limitations of the proposed framework, the authors further incorporate reinforcement learning and cosine alignment loss. Evaluation results demonstrate the effectiveness of the LlavaCode.

**Strengths:**

1. The studied problem is valuable. Indeed, code generation tasks suffer a lot from the long context problem in RAG, much more severe than general generation tasks.
2. The proposed LlavaCode demonstrates better performance on ES and EM.
3. The paper is well-written and easy to follow.

**Weaknesses:**

1. The proposed LlavaCode seems highly similar to the related work xRAG mentioned in the paper, which limits the novelty of this paper. The authors should explicitly discuss the distinction between xRAG and LlavaCode. I don't agree that xRAG was not evaluated under the code generation task is a solid reason. Besides, it seems that the authors do not compare the performance between xRAG and LlavaCode in the paper. If I ignore the comparison, please correct me.
2. Exact Match and Edit Similarity are not good metrics for evaluating code generation performance. Correct codes can have highly different linguistic forms due to variable naming preferences, different running logics, etc. This was largely discussed by previous works, which were referred to as style difference [1, 2]. Thus, recent code generation works often evaluate functional correctness through the pass rates of test cases, not the matching percentage of tokens.
3. The authors only evaluate the LlavaCode on Qwen-series LLMs and only on one dataset. More Experiments should be conducted to demonstrate the generalization ability of LlavaCode.

[1] Wang, Yanlin, et al. "Beyond functional correctness: Investigating coding style inconsistencies in large language models." Proceedings of the ACM on Software Engineering 2.FSE (2025): 690-712.

[2] Li, Haochen, Xin Zhou, and Zhiqi Shen. "Rewriting the Code: A Simple Method for Large Language Model Augmented Code Search." Proceedings of the 62nd Annual Meeting of the Association for Computational Linguistics (Volume 1: Long Papers). 2024.

**Questions:**

1. It would be interesting to investigate why the compressed representations become highly similar without cosine alignment loss. Does it mean that not all the retrieved texts are meaningful in RAG? I would like to hear some explanations from the authors.
2. The discussion of the limitations of CE loss confused me. The authors said that Exact Match, which measures the percentage of predictions character by character, is a sequence-level metric. I would like to hear some explanations from the authors.

---

> ### Author Response · Authors · 2025-11-21
>
> Thank you for your review. We are sorry to see that our work received a score of 2, and we appreciate the opportunity to clarify the concerns you raised.
>
> -  Distinctions Between xRAG and LlavaCode:
>
> In Section 3.2, we note that most prior work trains projectors exclusively with Cross-Entropy (CE) loss, with xRAG being the exception, using CE + KL. In Section 3.5, we evaluate this same CE-only and CE + KL settings and show that LlavaCode unique loss outperformed both.
> Our primary contribution is the development of a new, effective loss formulation for projector training in the code setting. This loss combines: of CE, an RL-based loss (Section 3.3) that, to our knowledge, has not been previously applied to projector training or compression tasks, and a novel cosine alignment loss (Section 3.4).
> To make our distinction from xRAG clearer, we now include CE + KL results (xRAG-style training) in Table 1 and also report results of the original publicly available xRAG model on code-completion tasks in Appendix B Table 5 .
>
> - Exact Match and Edit Similarity Metrics:
>
> To broaden the scope, we have included additional metrics such as CodeBLEU, along with experiments on larger models (see Table 2). CodeBLUE incorporates SyntaxMatchScore and DataflowMatchScore and thus can be more suitable for code.
>
> Considering the execution-based metrics: this is a tricky evaluation to perform, since code completion benchmarks often deal with uncompilable chunks of code (unlike function generation benchmarks), or consider repositories poorly covered by unit tests.
> We focused on EM and ES,  because these are the metrics that are very often reported in the literature on code completion. Additionally, EM can serve as a lower limit of the pass@1 execution metric.
>
> Nevertheless, we are in the process of compiling an appropriate benchmark to measure a pass@k metric and will try to add these results in a time before the deadline.
>
> -  Limited Evaluation Scope:
>
> We agree that broader evaluation is needed. Table 1 now strictly contains ablation studies, while results on CrossCodeEval, RepoEval, and RepoEvalAPI appear in Table 2, along with results on larger model.
>
> -  Meaningful Retrieved Context:
>
> Not all retrieved context is helpful in RAG settings. RepoBench, for example, includes an “in-file” subset specifically designed to test robustness to irrelevant retrieval. To further illustrate this issue, we added resilience metrics in Appendix B Figure 4, showing that retrieved context can harm correct predictions—and that our compression approach even slightly improves robustness to such irrelevant context.
>
> -  Limitations of CE Discussion:
>
> We agree that our phrase “character for character” in the EM definition may have caused ambiguity. We have revised it to clearly state that EM requires an exact match between generated and target sequences.
> Regarding why CE is token-level while EM is sequence-level:
> CE is computed by conditioning on the ground-truth tokens up to position n +t and evaluating the log-probability of token n+t+1. Summing over T tokens still yields a collection of token-level terms—each evaluated under teacher forcing—rather than a true sequence-level metric.
> In contrast, EM evaluates whether the entire generated sequence matches the target, with each predicted token conditioned only on the model’s own previous predictions. This distinction explains why CE cannot fully capture sequence-level correctness.

---

### Official Review · Reviewer_ndNz · 2025-10-28

**Soundness:** 2
**Presentation:** 2
**Contribution:** 1
**Rating:** 2
**Confidence:** 4

**Summary:**

This submission studies code representation compression for retrieval-augmented code generation (RAG). To support RAG for code completion, a contextual representation must be generated, which raises efficiency concerns. Prior work has explored context compression, among which embedding projection (e.g., LLaVA) is a typical approach. The current paper adapts LLaVA to code completion, resulting in LLaVACode. The authors evaluate their method using two metrics—Exact Match and Edit Similarity—and employ reinforcement learning (REINFORCE) to optimize these objectives.

**Strengths:**

- The idea of exploring compression techniques for code generation/completion is interesting and potentially useful for improving both efficiency and quality.

- The originality of the paper is low (cf. weakness).

- The paper is readable, partially because of the simplicity of the approach.

**Weaknesses:**

- Incremental contribution. The method is a direct adaptation of LLaVA. As shown in Figure 1(b), the encoder architecture is nearly identical to LLaVA. The adaptation process appears straightforward and does not deeply consider the unique characteristics of programming languages or the specific challenges of code generation compared with natural language. Although the paper incorporates AST information, this component is relatively weak and does not seem to yield notable performance gains.

- Inadequate evaluation metrics. The chosen metrics, Exact Match and Edit Similarity, do not faithfully capture the quality of code completion. While they may be suitable for natural language text, they are insufficient for code, where even minor syntactic differences can lead to uncompilable outputs. More meaningful metrics—such as functional correctness or execution-based accuracy—should be included to better assess code quality.

- Limited evaluation scope. The experiments are conducted exclusively on Qwen2.5-coder and limited to a Python dataset, leaving it unclear how the proposed approach generalizes to other models, languages, or programming paradigms.

**Questions:**

1. Why are Exact Match and Edit Similarity the only metrics considered?
2. Why choose REINFORCE? Why not choose state-of-the-art RL methods?
3. What is the essential novelty of LLaVACode compared to LLaVA?

---

> ### Author Response · Authors · 2025-11-21
>
> Thank you for your review. We are sorry to see that our work was rated 2.We hope we can clarify the raised questions and weaknesses you pointed out:
> -  Novelty:
>
> First, to clarify, the projector is a small, trainable MLP, similar to those used in prior work for mapping encoder outputs into LLM space. The encoder, by contrast, is a pre-trained model that remains unchanged and can have any architecture.
> While we do leverage ideas from xRAG (retrieved-context compression) and LLaVA-like projectors, our main contribution is twofold: i) Demonstrating that previously proposed projector training methods fail for code—a modality not previously studied in this context; ii) Designing a novel, complex loss that combines cross-enthropy, reinforcement learning techniques (previously unreported for compression or projection tasks) with a new cosine alignment loss.
> To highlight this, we updated Table 1 to include ablations using LLaVA (CE-only) and xRAG (CE+KL) loss functions, showing that LlavaCode outperforms both.
> Regarding the AST encoder: while it showed slightly lower ablation results, its strong performance relative to its smaller, older model size (compared to Qwen3Embedding) suggests the AST modality is beneficial. Unfortunately, alternative code modalities (AST, DFG, CFG) are not supported in newer code encoders, limiting direct comparisons.
>
> -  Evaluation Metrics and Scope:
>
> To expand our evaluation, Table 1 now shows ablation studies, and we provide benchmark results on CrossCodeEval, RepoEval, and RepoEvalAPI, along with experiments on larger models (see Table 2). To broaden our metrics, we have included CodeBLEU. CodeBLUE incorporates SyntaxMatchScore and DataflowMatchScore and thus can be more suitable for code.
>
> Considering the execution-based metrics: this is a tricky evaluation to perform, since code completion benchmarks often deal with uncompilable chunks of code (unlike function generation benchmarks), or consider repositories poorly covered by unit tests.
> We focused on EM and ES,  because these are the metrics that are very often reported in the literature on code completion. Additionally, EM can serve as a lower limit of the pass@1 execution metric.
>
> Nevertheless, we are in the process of compiling an appropriate benchmark to measure a pass@k metric and will try to add these results in a time before the deadline.
>
> - Why REINFORCE:
>
> Our approach trains only the projector, keeping the encoder and LLM frozen. This makes training lightweight and simple. Using PPO, A3C, or SAC would add complexity through value networks, critics, ratio clipping, KL penalties, and multiple epochs per batch.
> Moreover, since our reward (EM + ES) is deterministic and computed directly from the greedy generated sequence, we do not need variance reduction or advantage estimation. CE loss is sufficient to stabilize training, as we mention in Section 3.6.
> Our goal was a simple, computationally efficient training approach. We acknowledge that exploring PPO, GRPO, or other RL variants is interesting future work, as we discussed in the Conclusions section.

---

### Official Review · Reviewer_FAS8 · 2025-11-01

**Soundness:** 2
**Presentation:** 3
**Contribution:** 3
**Rating:** 6
**Confidence:** 3

**Summary:**

This paper proposes LlavaCode, a framework that compresses retrieved code context into compact embedding tokens for retrieval-augmented code completion. Inspired by LLaVA and xRAG, it uses a lightweight projector to align encoder embeddings (e.g., from Qwen3-Embedding or UniXCoder) with a frozen code LLM. The model is trained with a composite loss combining Cross-Entropy, REINFORCE (directly optimizing EM and ES), and Cosine Alignment to prevent projection collapse. Experiments on a self-constructed Python dataset show 20–38% latency reduction and small accuracy gains over a no-context baseline.

**Strengths:**

- Novel idea in a new domain: The first attempt to apply LLaVA-style projection and compression to the code RAG problem. This is conceptually creative and well-motivated by latency constraints in IDE-style code completion.

- Thoughtful loss design: The use of a composite CE + RL + cosine alignment objective is well-argued and clearly improves stability and alignment.

- Thorough ablation: The authors carefully isolate the effects of encoder choice, loss terms, and projection architecture, which adds credibility to the analysis.

- Clear motivation and solid writing: The methodology section is coherent and easy to follow, with intuitive visualizations (Figure 1–3) and detailed latency analyses.

**Weaknesses:**

- Limited effectiveness: Despite the clever design, LlavaCode’s generation quality is underwhelming. EM only rises from 45.97 → 47.66 (vs no-context), while full-context QwenCoder achieves 50.87. This small margin makes it hard to claim the approach “works” for code RAG in a meaningful way.

- Dataset limitation: All evaluations are conducted on a self-created Python dataset, not on established benchmarks such as CrossCodeEval or RepoEval. This leaves both generalization and fairness in doubt.

- Questionable compression validity: Code logic and dependency resolution are highly structured; compressing an entire retrieved snippet into a single embedding may lose critical syntactic and semantic cues. The current results seem to confirm this weakness.

- Scope of results: The experiments focus narrowly on EM/ES and TTFT. There is no human or task-level evaluation of completion usefulness or correctness, which matters for latency–quality trade-offs in IDEs.

- No comparison to advanced baselines: The paper compares only to “with” and “without context” Qwen baselines, missing broader comparisons to more sophisticated retrieval or context-pruning methods.

**Questions:**

- Why was evaluation restricted to self-constructed data? How would LlavaCode perform on standard repo-level benchmarks like CrossCodeEval or RepoEval?

- Could the authors show qualitative examples where compressed context helped or failed, to better illustrate interpretability?

---

> ### Author Response · Authors · 2025-11-21
>
> Thank you for your thoughtful review. We are pleased that you recognized the novelty of our approach and appreciated the complex loss design associated with applying context compression in the code domain.
> We have carefully addressed the weaknesses and questions you highlighted:
> Benchmarks:
> We agree that thorough comparisons on well-known benchmarks are essential. While Table 1 focuses on ablation studies, we have added results on CrossCodeEval, RepoEval, and RepoEvalAPI in Table 2.
>
>
> - Scope of Results:
>
> To broaden the scope, we have included additional metrics such as CodeBLEU, along with experiments on larger models (see Table 2). CodeBLUE incorporates SyntaxMatchScore and DataflowMatchScore and thus can be more suitable for code.
>
> Considering the execution-based metrics: this is a tricky evaluation to perform, since code completion benchmarks often deal with uncompilable chunks of code (unlike function generation benchmarks), or consider repositories poorly covered by unit tests.
> We focused on EM and ES,  because these are the metrics that are very often reported in the literature on code completion. Additionally, EM can serve as a lower limit of the pass@1 execution metric.
>
> Nevertheless, we are in the process of compiling an appropriate benchmark to measure a pass@k metric and will try to add these results in a time before the deadline.
>
> - Compression Validity:
>
>
> Code is indeed a highly structured modality. To account for this, we included the UnixCoder AST Encoder in our ablation studies. Although its results were slightly lower, given its smaller size and age relative to Qwen3Embedding, its relatively strong performance suggests that the AST modality was beneficial. Unfortunately, alternative code modalities (AST, DFG, CFG) are not supported in later code encoders, which limits direct comparisons.
>
> Also, CodeBLEU metric that we included in our results on benchmarks (Table 2) incorporates SyntaxMatchScore and DataflowMatchScore, connecting increase in CodeBLEU to the perseverance of syntactic cues.
>
> Additionally, some information loss during compression may sometimes be advantageous for retrieval-augmented generation: irrelevant retrieved context can harm model predictions. The xRAG paper and RepoBench benchmark highlighted this effect. Our compression even has a slightly better resilience rate than the uncompressed context approach (Figure 4 in Appendix B).
>
>
> - Qualitative Examples:
>
> As requested, we have included examples of code generations where LlavaCode performed better, worse, or on par with the model with uncompressed context in Appendix B Table 6.
>
>
> - Comparison to Advanced Baselines:
>
> We have added comparisons with alternative compression methods capable of achieving similar compression rates (Table 2).
>
>
> - Limited Effectiveness:
>
> Regarding the ES metric, our method performs even slightly better then uncompressed context-augmented models on some benchmarks (CCEvalLong, see Table 2).  While prediction quality may be lower on most benchmarks, our compression avoids the latency overhead of full context-augmented generation—essentially a “quality enhancement for free.”
>
> At the same time, methods that also could compress the cross-file context fail to bring any increase in the target metrics at the level of extreme compression achieved by LlavaCode (as shown on RepoEval in Table 2).

---

### Note · Authors · 2026-01-15

I have read and agree with the venue's withdrawal policy on behalf of myself and my co-authors.